# Combining Pharmacokinetics and Vibrational Spectroscopy: MCR-ALS Hard-and-Soft Modelling of Drug Uptake In Vitro Using Tailored Kinetic Constraints

**DOI:** 10.3390/cells11091555

**Published:** 2022-05-05

**Authors:** David Pérez-Guaita, Guillermo Quintás, Zeineb Farhane, Romá Tauler, Hugh J. Byrne

**Affiliations:** 1FOCAS Research Institute, Technological University Dublin, City Campus, D08 CKP1 Dublin, Ireland; zeafarhane@gmail.com; 2Department of Anaytical Chemistry, University of Valencia, 46100 Valencia, Spain; 3Health and Biomedicine, Leitat Technological Centre, 08028 Barcelona, Spain; gquintas@leitat.org; 4Institute of Environmental Assessment and Water Research (IDAEA)—Higher Council for Scientific Research (CSIC), 08043 Barcelona, Spain; roma.tauler@idaea.csic.es

**Keywords:** Multivariate Curve Resolution-Alternating Least Squares, pharmacokinetics, Raman microspectroscopy, chemometrics

## Abstract

Raman microspectroscopy is a label-free technique which is very suited for the investigation of pharmacokinetics of cellular uptake, mechanisms of interaction, and efficacies of drugs in vitro. However, the complexity of the spectra makes the identification of spectral patterns associated with the drug and subsequent cellular responses difficult. Indeed, multivariate methods that relate spectral features to the inoculation time do not normally take into account the kinetics involved, and important theoretical information which could assist in the elucidation of the relevant spectral signatures is excluded. Here, we propose the integration of kinetic equations in the modelling of drug uptake and subsequent cellular responses using Multivariate Curve Resolution-Alternating Least Squares (MCR-ALS) and tailored kinetic constraints, based on a system of ordinary differential equations. Advantages of and challenges to the methodology were evaluated using simulated Raman spectral data sets and real Raman spectra acquired from A549 and Calu-1 human lung cells inoculated with doxorubicin, in vitro. The results suggest a dependency of the outcome on the system of equations used, and the importance of the temporal resolution of the data set to enable the use of complex equations. Nevertheless, the use of tailored kinetic constraints during MCR-ALS allowed a more comprehensive modelling of the system, enabling the elucidation of not only the time-dependent concentration profiles and spectral features of the drug binding and cellular responses, but also an accurate computation of the kinetic constants.

## 1. Introduction

The development of microscopy in the 16th/17th centuries has changed our fundamental understanding of the world in which we live, revealing the complex structures of the cellular building blocks of life itself. The emergence of labelling techniques in fluorescence microscopy in the early 1900s [1] has enabled the elucidation of key biochemical processes at a subcellular level. Technological developments such as confocality, laser rastering, resonance transfer, fluorescence lifetime, and more recently, super resolution imaging have pushed the spatial resolution ever lower, such that microscopic imaging techniques have become an essential tool in in vitro biological research, as well as in drug discovery and development, and toxicology. However, there is a dearth of methodologies to visualize the intracellular interactions of small molecules, critical to medicinal chemistry, and the ability to visualize specific processes in cells is dependent on the introduction of colorimetric labels, such as fluorescent tags of biomolecules or organelles, which act as sentinels of the specific processes identified, and only those processes. Notably, the labelling process assumes an a priori knowledge of the pathway to be labelled, and monitoring the full range of responses of a cell or cell population to an external stimulus requires multiple labels and sources, as well as exhaustive replicate monitoring over multiple time-points and exposure doses [2,3,4]. The cost implications for fundamental academic research are limiting [5], and the limited information gleaned even from multimillion, automated High Content Analysis (HCA) installations has restricted the exploitation of in vitro models at the preclinical screening stage of drug development. Improved in vitro screening methods to increase the speed and reduce cost of analysis are therefore highly desirable [6].

Vibrational spectroscopic microscopy (microspectroscopy) has emerged as a label-free alternative, which can provide molecularly specific signatures of biological processes and function. In particular, Raman microspectroscopy can be performed at visible wavelengths, and in a confocal mode can provide molecularly specific detail at a subcellular level, of live or fixed cells, in a label-free fashion [7,8]. In vitro spectroscopic studies of cells allow a detailed analysis of the fundamental cell biology, or biochemical changes, for example, as a result of an external agonist, toxicant or chemotherapeutic agent [9,10,11], promising potential applications in fundamental cellular (cytological) research, medicinal chemistry, and pharmacological/toxicological screening.

However, the analysis and the extraction of useful, actionable, information from the data generated during label-free spectroscopic analyses remains challenging. In the absence of labelled biomarkers, it is not trivial to interpret and represent the observations in terms of biological processes. Due to the complexity of the biochemical milieu of cells or tissue, spectral responses are convoluted, changes can be subtle and multivariate, and analysis relies on an arsenal of multivariate chemometric techniques [12]. In the case of diagnostic applications, classification methods, such as clustering or principal components analysis (PCA), are commonly used. More sophisticated techniques, such as multivariate linear regression analysis, have been employed, for example, for quantitative analysis of human blood serum [13], and to explore the aetiology of disease [14], correlating the differential evolution of spectral changes with a target variable, such as viral copy number per cell or radiation dose [15,16]. In the case of chemotherapeutic agents, regression analysis such as partial least squares regression (PLSR) has been employed, for example, to independently elucidate the spectroscopic signatures of the direct chemical effects of the stimulus from the subsequent cellular metabolic responses [17,18]. In linear methods such as PLSR a constant response is assumed between the predictors (i.e., spectral variables) and the dependent variable (i.e., time). However, the variation of biochemical composition within the cell follows complex non-linear patterns established by the kinetic relationships between the different players (e.g., drugs, metabolites, nucleic acids, cell responses) which cannot be described using linear models. 

Multivariate curve resolution-alternating least squares (MCR-ALS) has been proposed as a more flexible approach to study cell uptake processes [19]. This methods allows the resolution of chemical mixtures in time-dependent experiments and provides information about the evolution of the chemical components, but requires the input of accurate information about the system in order to avoid ambiguities. 

A particular feature of MCR-ALS is the use of external information (e.g., initial estimations of component concentrations or spectra) and constraints to guide the iterative process by reducing the set of possible solutions of the bilinear curve resolution decomposition and consequently reducing the ambiguity of the results, leading to chemically interpretable solutions. One of these constraints involves the implementation of a hard-and-soft approach which fits the evolution of the concentration profiles of the components to known chemical kinetic reactions defined by rate constants. This method has been used to study kinetic equations in the monitoring of simple reactions (e.g., A → B, A → B → C, A + B → C + D). In this paper, we introduce a phenomenological rate equation approach to describe the kinetics of the drug uptake by, and binding within biological cells, as well as their cellular responses. This system is defined by a set of ordinary differential equations which can be implemented as a kinetic constraints during an iterative optimization process in the MCR-ALS method. This strategy offers the possibility to data-mine the kinetic evolution of the characteristic signatures of cellular uptake of the drug, its subcellular interactions, and the subsequent cellular responses. The methodology is first demonstrated using simulated data and then applied to empirical data from previous experiments concerning the inoculation of A459 and Calu-1 human lung cells with doxorubicin (DOX), in vitro. 

## 2. Materials and Methods

### 2.1. Hard-and-Soft MCR-ALS Modelling

MCR-ALS models the dataset of cell spectra measured at different inoculation times by considering a set of components whose concentrations evolve at different rates and have characteristic spectral features. MCR-ALS iteratively solves the following bilinear equation,
(1)$$D=CS^{T},$$
where the experimental data matrix **D** (**N** × **J**) is decomposed in a pure spectral matrix **S** (**I** × **J**) and a concentration matrix **C** (**N** × **I**), where **N** is the number of spectra, **J** the number of spectral variables, and **I** the number of components. 

The scheme depicted in Figure 1a shows the iterative process underpinning MCR-ALS. Herein, we include a short description to describe the inclusion of the pharmacokinetic equations within the modelling, but more detailed information can be found in selected references [19,20]. Firstly, the number of components to be modelled is established by considering previous or external information of the system or by using techniques such as Singular Value Decomposition (SVD) [21]. Then, an initial estimation of concentration matrix or spectra (in Figure 1a, we have used the concentration) is calculated by using information about the chemical components involved or using other methods such as Evolving Factor Analysis (EFA) [22]. The estimated concentration matrix is then constrained using criteria which depend on the known features of the signals and components (i.e., non-negativity) and then the dataset is used to obtain an estimated spectral matrix. This estimated pure spectral matrix is constrained again (for example, eliminating possible negative bands) and combined with the estimated concentration matrix to compute an estimated dataset, which is compared with the actual one to calculate a fitting error. If this error satisfies an established convergence condition, the iteration is stopped, or if not, the empirical dataset and the constrained pure spectra are used to create a revised estimated matrix of concentrations which is again introduced in the optimization process until convergence. In a similar way to previous studies [23,24], the convergence was achieved when the fitting error, measured as the sum of the squares of the residuals of the reconstructed matrix with respect to the original one, increased or decreased less than 1% for 20 iterations or the process reached a maximum of 200 iterations.

In this study, we incorporate a hard-and-soft approach by introducing kinetic constraints (Figure 1b) within the iterative process of MCR-ALS. For this purpose, a system of differential equations describing the kinetic uptake of the drug as well as the corresponding cellular responses, containing the relative concentration of the components and their kinetic constants (k1, k2…k), is considered. When this system is applied as a constraint, the kinetic constants of the system at each MCR-ALS iteration are fitted using the *lsqcurvefit* non-linear least squares solver of MATLAB (Mathworks, Natick, MA, USA), updating the previous estimation of the constants ($k_{0}$) at the relevant time points. The best estimated values of $k_{s}$ are then used to calculate the constrained concentrations in the next iteration of the MCR-ALS optimization process. Therefore, at each iteration, the proposed approach includes the optimization of the concentration and spectral factor matrices and, in addition, the optimization of the constants of the kinetic equations as shown in the next workflow.

### 2.2. Data Analysis

Data analysis was carried out in MATLAB R2021a using in-house written functions based on modified functions of the MCR-ALS toolbox (https://mcrals.wordpress.com/download/mcr-als-2-0-toolbox/, accessed on 1 October 2021) and incorporating the *lsqcurvefit* function from the MATLAB Optimization toolbox. The Calu-1 and A459 experimental datasets were obtained from, and the simulated datasets calculated on the basis of, previous studies, using a time interval of 0.5 h over the range between 0 and 72 h after inoculation [25,26]. Functions and scripts as well and datasets used are available in the Zenodo repository (10.5281/zenodo.6385490). In order to start the MCR-ALS analysis, an initial guess of the concentration profiles (C0) was calculated by solving ordinary differential equations (ODEs) at pre-established initial kinetic constant (*k_n_*) values. These initial guesses of the *k_n_* values can be selected arbitrarily or by using known information about the kinetics of the system. Non negativity constraints were introduced for both concentration and spectral matrix, and the initial datasets were normalized using the area of the phenylalanine band obtained by integrating the 988–1023 cm^−1^ using a baseline fitted in the 986–990 cm^−1^ and 1021–1025 cm^−1^ regions.

## 3. Results

### 3.1. Simulated Datasets

To initially demonstrate the validity of the approach, we used simulated data. Following a previous model [24], we considered the inoculation of A549 cells by a single dose of DOX (D). We simulated the uptake $N_{b}$ and cell responses $N_{r}$ using the following two differential equations;
(2)$$\frac{dN_{b}}{dt}=\left(N_{recp}-N_{b}\right)K_{up}D,$$
(3)$$\frac{\;dN_{r}}{dt}=\left(N_{resp}-N_{r}\right)K_{resp}N_{b},$$
where $K_{up}$ is the uptake rate of the drug into the cell, $K_{resp}$ is the response rate, $N_{recp}$ is the number of drug receptors, and the $N_{resp}$ is a measure of the cellular response. The two latter constants act as limiting factors to the drug uptake and response observed experimentally.

The constants were chosen to describe a typical rapid uptake followed by a saturation, as has been experimentally observed (Figure 2a) [25]. The associated spectral changes, characteristic of the drug binding, are depicted in cyan in Figure 2b, and include typical bands from DOX (460, 440, and 1211 cm^−1^), the spectrum of which is shown in dark blue in Figure 2b, as well as changes in the RNA/DNA bands caused by the drug binding (785 and 811 cm^−1^). In order to simulate the observed effect of the binding on the RNA/DNA bands, manifest as a decrease of the Raman intensity, the “binding” spectrum was subtracted rather than added. Equation (3) describes the response of the cell, which is slower than the initial drug uptake and binding, as it is produced as a result of the drug binding to RNA/DNA (Figure 2a). The associated spectrum, shown as green in Figure 2b, was created to represent changes in the protein bands (1450 cm^−1^). A simulated dataset of 145 spectra was created by adding the spectral contributions of the drug uptake and of their RNA/DNA binding multiplied by the concentration changes calculated by resolving the Equations (1) and (2) between 0 and 72 h. A total of 1% of noise was added using the *rand* function from Matlab.

This simulated dataset was then analyzed using the proposed hard-and-soft MCR-ALS method. The goal of this data analysis was to assess whether, under the assumption of known kinetics, MCR-ALS can resolve the correct spectral and concentration profiles, as well as the rate constants associated with the proposed kinetic equations. A set of initial values for the kinetic constants were arbitrarily selected as initial conditions (Figure 2f), being values very different from the ones used in the simulation. Three components were used, considering results from SVD, which is a rank method used to establish the number of components in MCR-ALS. In short, it shows the percentage of variance captured by the different components, and the user selects the minimum number of components which explain considerable variance, ensuring that components related just to pure noise are not selected. The three components selected in principle represent the two variable effects (responses) introduced and the aspects of the cellular spectra which are not affected by the drug uptake. The latter could have been eliminated by subtracting the spectrum of the untreated, control cell from the entire the dataset. However, considering that one of the effects of the drug is a reduction of the DNA/RNA bands, ref. [25] the initial component spectra has more intense bands than the cell spectra inoculated with the drugs. This could have resulted in undesirable negative peaks caused by negative contributions of the RNA/DNA interactions, so we decided to not subtract the unaffected cell spectra and include it as a component on the MRC-ALS model. Figure 2c shows the predicted kinetic evolution of the component concentrations, while Figure 2d shows their spectral profiles, after 200 iterations. The kinetic evolution matches well that of the simulated dataset (Figure 2a), and notably, Component 3 (orange), which matches well the spectrum of the untreated cell, is constant over time, as expected. The other components showed clear similarities with the simulated effects. Component 1 (blue) showed a rapid evolution, similar to the simulated drug binding (Figure 2a), and its spectral profile matches well that of the combination of the drug and the negative bands associated with the binding (Figure 2c). Component 2 (green) shows a slower evolution, similar to the simulated response (Figure 2a), and the spectral profiles accurately reproduce the simulated response (Figure 2d). 

In addition, the model provided an estimation of the constants of differential Equations (2) and (3), which are shown in Figure 2e. With each iteration, the estimation of the rate constants, *K_up_* and *K_resp_*, evolved from their initial input value (with units of h^−1^), converging to accurately reproduce the value used in the simulation (Figure 2f). In the case of the constants *N_recp_* and *N_resp_*, quantifying the amount of uptake and cellular response, the estimated values also converge, although their absolute values differ from the theoretical ones due to the spectra normalization used during the ALS optimization. This intensity ambiguity in the units of the estimated constants can only be resolved by calibrating the strength of the Raman spectral response with the amount of substance measured (concentrations). In summary, the analysis of this simulated dataset indicates that, based on an appropriate set of equations describing the kinetics of the system the hard-and-soft MCR-ALS modelling of the spectral data is capable of extracting the information about the cellular drug uptake process, including the spectral and concentration profiles, as well as the rate constants.

### 3.2. Real Dataset 1: Study of DOX Uptake by A549 Cells In Vitro Using Raman Microspectroscopy

Next, we tested the proposed approach on real data using results of a previous experiment in which the uptake of DOX by A549 cells in vitro was monitored in different compartments of the cell, the nucleolus, nuclei and cytoplasm, using Raman microspectroscopy (Figure 3a) [25,27]. The main goal was to establish how the hard-and-soft MCR-ALS method can be used to investigate differences in the uptake rates in the different cellular regions, so each dataset for the nucleolus, nuclei and cytoplasm was treated independently. For this purpose, two components were modelled initially, for each cellular region. One was left unconstrained, while the second one was constrained through the hard-and-soft model approach by using Equation (2), with the same initial estimates used in the previous section.

The result of the three different hard-and-soft models can be seen in Figure 3b–g. For each cellular region, the modelling identified two components, the first one (Component 1-green) was the one not affected by the constraints. In all cases, this component was relatively constant over time (Figure 3c), and its spectrum represents a typical cellular profile, featuring, for example, the phenalanine and amide III and I bands found at 1005, 1490, and 1640 cm^−1^, respectively. In general, the pure spectra of the three systems show similar bands (see Appendix A for a full comparison), except for nucleic acid bands found in the 750–850 cm^−1^ region, which show larger intensities in the nucleus and nucleolus. In contrast, the concentrations of the second component, shown as blue in Figure 3b–d, evolved significantly with time. The concentration profile for all the compartments increased, albeit at different rates. In the nucleolus, it increased sharply, reaching a plateau within ~10 h. For the nucleus, the increase was slower, and the plateau was reached only after ~40 h. In the case of the cytoplasm, however, the increase was so slow that the saturation was not reached within the 72 h time span of the experiment.

In all cellular regions, the spectrum of the second (Component 2-blue) clearly shows bands from DOX at 460, 440, and 1211 cm^−1^. The spectral and concentration profiles therefore indicate that the constrained component incudes the signatures of the increasing uptake of DOX in the different cellular regions as a function of time. Furthermore, in the case of the nucleolus, negative features of the nucleic acids bands at 785 and 811 cm^−1^ are observed, indicating that Component 2 also captures changes in cellular features caused by the rapid binding of DOX to the biomolecules, including the decreasing of the intensity if DNA/RNA bands. In comparison, the spectra for the DOX components in this wavenumber region shows no absorption bands for the cytoplasm and positive bands for the nucleus, indicating the normal intensity of unbound nucleic acids. Most importantly, the proposed approach could estimate the kinetic uptake constants ($K_{up}$ in Equation (2)) for the three components, being 0.25, 0.16, and 0.075 h^−1^ for nucleoli, nucleus, and cytoplasm, respectively. This quantifies the changes in accumulation rates for the three cellular regions, and the results are consistent with the experimental observation of rapid accumulation and saturation initially in the nucleoli of the cell, followed by the nucleus and finally cytoplasm [25,28].

In an attempt to extract the subsequent evolution of the cellular response, three components were also considered in the model, corresponding to the initial cell spectrum, the drug uptake and binding, and the subsequent response, described by Equations (2) and (3). In a similar way to the simulated data, we used Equations (2) and (3) as input to define the kinetic constraints, with the same initial estimate for the constants. After the first iteration, the modelling confused the binding and subsequent responses, making the kinetics of the response faster than the uptake. Results of this experiment indicated that MCR-ALS was not able to extract information of the response from the experimental Raman dataset. Notably, however, the experimental results only contained data points at discrete times of 1, 2, 4, 6, 12, 24, 48, and 72 h, in contrast to the simulated dataset, which had data at 0. Five-hour intervals, and therefore, although the short-term evolution could be well modelled, the performance for the longer-term responses is not satisfactory. The experiment was designed with high temporal resolution in the first 24 h of inoculation to monitor the uptake and binding in detail. The secondary response, whose impact is more notable after the binding of the cell, was more difficult to track over the prolonged timescales. 

### 3.3. Real Dataset 2: Study of DOX Uptake by Calu-1 Cells

A second real dataset was employed to further test the proposed approach, which was composed by spectra of cytoplasm, nucleus and nucleolus in an incubation experiment performed in Calu-1 cells with DOX. Similar to the A459 experiment, two components were modelled. The first was constrained to represent the uptake and the second one was left unconstrained to model the cell components unaffected by the drug. Figure 4a depicts the calculated spectra of the kinetically constrained component, showing again the typical spectra of DOX. The nucleolus spectra also show the negative bands of the nucleic bands in the 750–850 cm^−1^ region, indicating, once again, the interaction of the drug with DNA and RNA. 

In this case, however, the computed profiles of this component (see Figure 4c) indicate a faster saturation in the nucleus than in the nucleolus, with also a very slow uptake in the cytoplasm. These uptake values of 0.007, 0.092, and 0.015 h^−1^ for cytoplasm, nucleus and nucleolus, respectively, indicate a different uptake behavior compared with the A549 cells. Finally, the second cell component spectra depicted in Figure 4b shows the typical spectra of biological compounds and their kinetic evolution for all components (Figure 4d) is constant with time.

## 4. Discussion 

Cellular physiology is a complex, dynamic system of highly synergistic processes, which is poorly represented by static omics models, based on a limited set of biomarkers. However, prompted by the need for a more in depth and integrated understanding of whole cell function, fundamental to strategies underpinning human healthcare, nutrition, and pharmacology, approaches to representing and understanding the mammalian cell metabolism based on kinetic physicochemical models are becoming increasingly sophisticated and popular [29,30]. 

Identifying key biomarkers in such pathways is critical, however, and conventional approaches can be blinkered by a priori assumptions. In contrast, label-free approaches such as confocal Raman microspectroscopy can provide a holistic, real-time representation of the biochemistry of the whole cell, at subcellular levels, and has previously been employed for characterization of the biochemical processes during cell culture and mitosis [31,32], proliferation [33], differentiation and activation [34,35,36], adhesion [37], death [38], and invasion [39]. Subcellular screening of drug uptake and mechanisms of action [10,25,40,41,42], and nanoparticle toxicity [9,43,44,45,46], in multiple cell lines have demonstrated a remarkable reproducibility of the subcellular signatures, suggesting a “spectralomics” approach to label-free characterization of cellular processes according to characteristic spectroscopic signatures is feasible. In this context, accurately data mining the characteristic signatures of the subcellular interactions and processes represents a challenge. Perez-Guaita et al. have previously demonstrated the application of MCR-ALS to gain insight into the pharmacodynamics and biochemical changes associated with drug exposure in an in vitro cellular model, as measured using Raman microspectroscopy [23], and a multimodal combination of Raman and Infrared microspectroscopies [24]. Similarly, making use of simulated datasets, and guided by kinetic constraints, the MCR-ALS method was demonstrated to be able to extract rather accurately the characteristic signatures of the drug uptake, intracellular binding, and subsequent cellular response. 

The current study demonstrates that the methodology can be more precisely constrained using a phenomenological rates equation approach describing the kinetics of the system. Accurate quantitative evaluations of key rates can be then employed to characterize the process, and, for example, to compare the responses of two different cell lines. The comparison of the performance of the proposed MCR-ALS method in the analysis of simulated and experimental data has, however, highlighted the importance of having sufficiently rich time course measurement datasets.

In general, the trend of results matches the previous experimental observations, although it should be noted that the previous analysis was of the principal component loadings which differentiated the spectra at a fixed timepoint from those of the unexposed control [25]. The kinetic analysis differentiates the trafficking of the drug in the two cell lines, indicating an initial accumulation in the nucleolus of A549 cells, whereas in Calu-1 cells, the nucleolar uptake is somewhat slower. Such kinetic analysis may be key to understanding, for example, the differences in cytotoxicity responses of the two cell lines to DOX [27,28], and similar rate equation approaches have been applied to nanoparticle [47,48] and drug uptake [49] by cell populations, and subsequent cellular responses, guided by classical toxicological analyses, faithfully reproducing a range of cytotoxicological response paradigms [50]. The emergence of open source resources such as Cell Designer [51], which enable biochemical pathway models to be simply drawn, but also kinetically modelled by the user, have made the approaches much more accessible. Model/Graphical representation by Systems Biology Markup Language (SBML) [52] and Systems Biology Graphical Notation (SBGN) [53], respectively, means that the user defined models can be interfaced with many other databases, for pathway enrichment [54], a technique becoming increasingly popular in predictive toxicology [55]. 

Results obtained for both the simulated and real datasets also point to some of the limitations of the hard soft MCR-ALS approach. Firstly, the secondary response, which was accurately tracked in the simulated dataset, was not found in the empirical datasets, probably due to the low temporal resolution used in the later stages of the experiment. This indicates that the technique only works when adequate timepoints representing the kinetic process of interest are measured. This issue relates to the challenges of preparing and measuring multiple replicates of inoculated and fixed cells over prolonged timescales, and can be addressed with the introduction of new live measurement techniques as well as more rapid Raman instruments.

Another important limitation is the specificity of Raman for some molecules. The Raman spectra only represent a limited number of biomolecules which provide intensities significantly above the noise. Although these biocomponents can be modelled in the kinetic model within the hard-and-soft approach, they may introduce ambiguity which reduces the capacity to identify and track the relevant components. Thus, in principle only biological processes with a significant number of components represented in the Raman spectra can be modelled. Furthermore, when increasing the complexity and number of the reactions involved in the process, the number of kinetic constants to model also increases, increasing the risk of obtaining ambiguous solutions by exchanging constants. 

## 5. Conclusions

In conclusion, a framework of label-free, subcellular Raman microspectroscopic analysis, combined with a kinetic, mechanistic modelling approach, to underpin chemometric analysis protocols, may provide a basis for the unambiguous interpretation of the evolution of the characteristic spectroscopic signatures. The approach lays the foundation for a spectralomics paradigm of label-free high content spectroscopic analysis technique for analysis of cellular function, providing a holistic view of the cellular processes to augment conventional labelled and omics approaches. 

## Figures and Tables

**Figure 1 cells-11-01555-f001:**
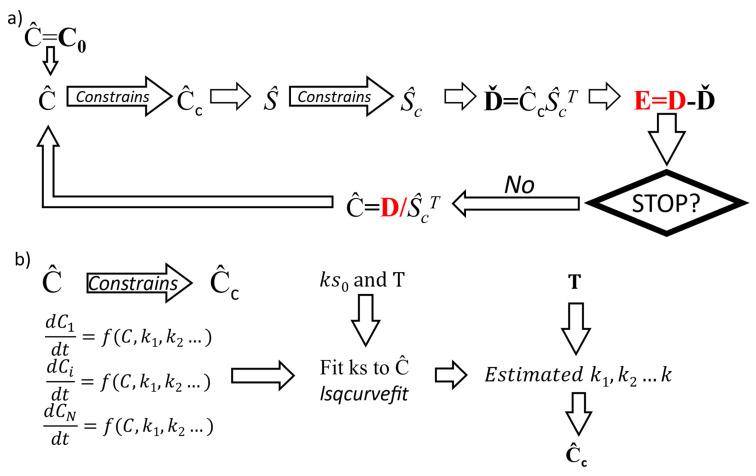
Workflow of (**a**) the MCR-ALS and (**b**) the hard-and-soft proposed.

**Figure 2 cells-11-01555-f002:**
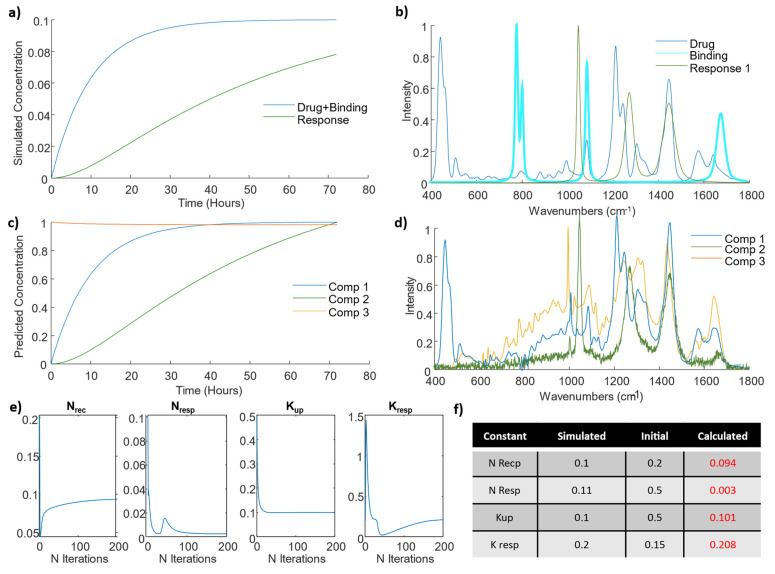
(**a**) Kinetic evolution of simulated drug uptake and binding (dark blue), and cellular response (green), (**b**) Raman spectrum of the drug, doxorubicin (dark blue), simulated spectral signature of drug binding (cyan) and of the subsequent cellular response (green), (**c**) predicted kinetic evolution of the MCR-ALS components, (**d**) MCR-ALS components, extracted after 50 iterations, (**e**) evolution of the kinetic constraint constants over 200 iterations of the MCR-ALS algorithm, (**f**) initial and final (after 200 iterations) constants employed in the MCR-ALS model.

**Figure 3 cells-11-01555-f003:**
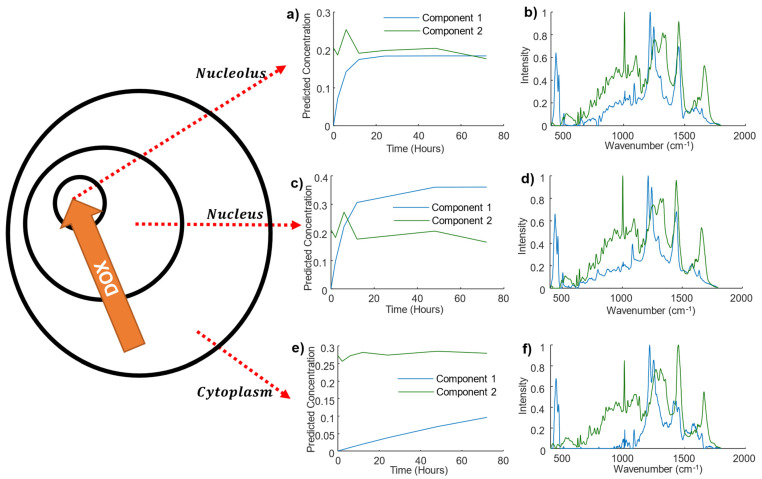
Schematic representation of hard-and-soft MCR-ALS analysis of Raman spectra of different cellular compartments of A459 cells incubated with DOX. Calculated concentration and pure spectra for the nucleoli (**a**,**b**), nucleus (**c**,**d**), and cytoplasm (**e**,**f**).

**Figure 4 cells-11-01555-f004:**
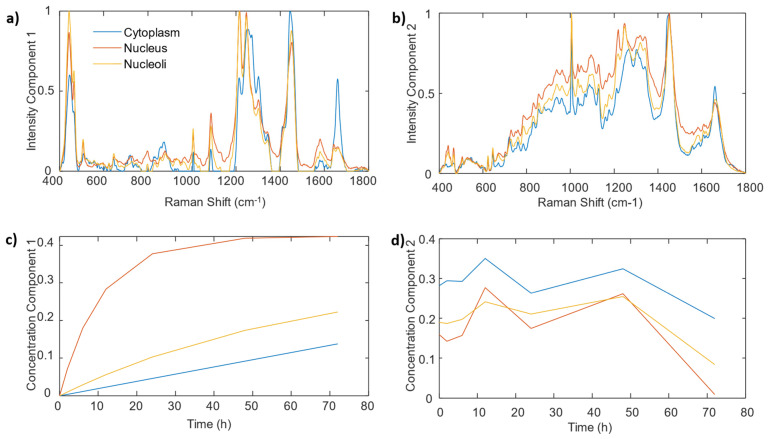
Hard-and-soft MCR-ALS analysis of Raman spectra from Calu-1 cells incubated with DOX. Calculated spectra for component 1 (**a**) and 2 (**b**). Concentration profiles obtained for component 1 (**c**) and 2 (**d**).

## Data Availability

Functions and scripts as well and datasets used are available in the Zenodo repository (10.5281/zenodo.6385490).

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
