# Peer review of "Combining Pharmacokinetics and Vibrational Spectroscopy: MCR-ALS Hard-and-Soft Modelling of Drug Uptake In Vitro Using Tailored Kinetic Constraints"

_cells, 2022, doi:10.3390/cells11091555_

Round 1

Reviewer 1 Report

The manuscript cells-1680078 proposed the integration of kinetic equations in the modelling of drug uptake and subsequent cellular responses using Multivariate Curve Resolution-Alternating Least Squares (MCR-ALS) and tailored kinetic constraints, for the analysis of Raman spectral data.

Possibilities of the proposed method have been clearly presented and discussed, however, a proper discussion about the limitations is missing.  I think this would complete and rather improve the manuscript.

Please correct typos in Figure 3 ("nuceloli", "nucelous")

Reviewer 2 Report

The authors presented a study on the integration of kinetic equations and MCR-ALS for the extration of chemometric features from Raman spectra of cells exposed to drugs.

The content of the article might be of interest to the readers of Cells but before being considered for publication the following points should be addressed. Furthermore, the manuscript should be thoroughly checked for typos, mispelling, and grammar mistakes.

Introduction

  • The section is confusing and not particularly well and logically organised
  • Line 43: Medicical chemistry should be changed into Medical chemistry
  • Lines 44 -  50: Relevant references must be included
  • Lines 51 -  54: Relevant examples and references must be included
  • Line 58: Spectroscopic microscopy is redundant. Either spectroscopic or miscroscopy
  • Line 60: Sentence "can be performed ... a confocal mode" is unclear and must be rephrased
  • Line 68: How does "interaction with light with complex biological systems" relate to what said beforehand?
  • Line 69: When the authors refer to "high content data" what do they mean? What type of content? Explain further
  • Line 70: Sentence "label free ... and the challenge" is unclear
  • Line 72: What milieu do the authors refer to? What type of complexity does this milieu have? Explain further
  • Lines 87 - 91: The authors explain the limitations of PLSR but they do not clearly explain the limitations of other multivariate techniques

Materials and Methods

  • Line 134: How is the convergence condition established? Explain further
  • Line 160: It is either "obtained" or "calculated"
  • Line 161: How were the kn values pre-established? Explain further
  • Line 163: Why were the initial datasets normalised? Why was phenylalanine chosen for normalisation? Explain further
  • Line 164: The whole sentence is unclear
  • Line 178: Sentence "the uptake ... equation 2" is a repetition
  • Line 178: Are the "constants" Nrecp and Nresp? Clarify in manuscript
  • Line 181: It would be better to avoid colours and to use different types of lines instead (e.g. dotted lines, dashed lines) to avoid confusions. This comment applies to all the graphs in the manuscript
  • Line 186: It is "slower" compared to what?
  • Line 204: How were "different initial values" chosen?
  • Line 204: Figure 2f is a table and it should be considered as such, i.e. labelled as a table and not as a figure
  • Line 205: Explain further how results from SVD were used to choose the three components
  • Line 207: Are the cells "not affected by the drug uptake" the control? Clarify in manuscript
  • Line 208: Are "untreated cells" the control? Clarify in manuscript
  • Line 209: What do the authors mean by "negative peaks"? Do they mean a decrease in the intensity of the peaks? Clarify in manuscript
  • Line 209: Are the "negative contributions of RNA/DNA interactions" the only ones considered?
  • Line 211: It is uncleat what "respectively" refers to in this context
  • Line 211: In the caption of the corresponding figure it says 50 iterations and not 200 as in the text
  • Lines 213 - 219: It is unclear what graphs have been considered. Furthermore, it is not sufficiently clear what components 1, 2 , and 3 are. Further explanation is needed
  • Line 225: What is the error within which the convergence occur? Clarify in manuscript
  • Lines 225 - 226: Clarify why normalisation was carried out on spectra since it introduced a discrepancy between experimental and theoreticcal values
  • Line 227: Why "units" and not values?
  • Line 228: What "externally calibration information" is that?
  • Line 230: Is not obvious that only if the most appropriate equations are used the best and most reliable results are obtained?
  • Line 248: This should be Fig. 3c and not Fig. 3b-d
  • Line 253: What are these "compartments"? Do the authors mean components?
  • Line 257: How long was the "time-span"?
  • Line 258: What "component 2" is that?
  • Line 262: What do "negative features" represent? A decrease in intensity?
  • Line 265: What do "positive bands" represent?
  • Fig. 3: It is not clear what the drawing represents. What do "R1" and "R2" stand for? Nucleoli and Nucleous are mispelled. There are no labels for the Raman spectra. Were the spectra normalised to 1? With respect to which peak?
  • Line 276: It is either "further" or "subsequent"
  • Lines 279 -  282: What was this negative result due to? Further explanation needed in the manuscript
  • Line 283: How does that related to the results presented earlier?
  • Line 286: How could the performance be improved over longer periods of time? Why such a negative result occur?
  • Line 296: What does "intercalation of the drug with DNA and RNA" mean?
  • Figure 4: Graphs are not labelled (i.e. labels (a), (b), etc. are missing). Legends are missing so that it is unclear what each spectrum refers to. Why are components 1 and 2 kept separated? There are some typos

Discussion and conclusion

  • This section is another introduction rather than a proper and exhaustive discussion of the results presented in the previous section. No clear conclusions are drawn. This section should be rewritten so that it critically ancd clearly discuss the results obtained.

Round 2

Reviewer 2 Report

Thank you to the authors for taking on board the comments. The section Introduction is still too long but it is clear that you have worked further on it. The section Discussion has been merged with Conclusion. This is quite unusual in a research paper and does not seem to have added any clarity to the article. Thank you for working a bit more on the section Discussion. It looks a bit more pertinent. 

Author Response

The authors tried their best to amend the paper. The section Introduction is still too long and some of the information is either irrelevant or redundant.

The authors feel that the introduction describes the context of the importance of cell microscopy to biological research and drug discovery, while also highlighting the limitations of current, commonly employed methodologies, based on extrinsic labels. It argues the case for label free approaches, such as vibrational spectroscopy, but also the requirement for more sophisticated datamining techniques, particularly to extract valuable information about the kinetics of cellular processes.

The authors have attempted to further reduce the Introduction, without losing the logistics of the context of and justification for the study. In particular, some introductory material (including Equation (1) has been moved to the Materials and Methods section.

The section Discussion has been improved slightly but has now been merged with the section Conclusions. This is not only unusual but also unnecessary. It would have been better to keep them separate.

The Original submission had the same structure, finishing with a section entitled “4. Discussion and Conclusion” and the structure was not changed in the revision process. This was informed by the Guide to Authors, but also by the fact that conclusions are drawn about different aspects of the study as they are discussed. We felt that reiterating them in a Conclusions section would add unnecessary repetition:

  1. Discussion

Authors should discuss the results and how they can be interpreted from the perspective of previous studies and of the working hypotheses. The findings and their implications should be discussed in the broadest context possible. Future research directions may also be highlighted.

  1. Conclusions

This section is not mandatory but can be added to the manuscript if the discussion is unusually long or complex.

On advice of the reviewer, the last paragraph of the section has been separated into a new section 5. Conclusions:

5. Conclusions

In conclusion, a framework of label free, subcellular Raman microspectroscopic analysis, combined with a kinetic, mechanistic modelling approach, to underpin chemometric analysis protocols, may provide a basis for the unambiguous interpretation of the evolution of the characteristic spectroscopic signatures. The approach lays the foundation for a spectralomics paradigm of label free high content spectroscopic analysis technique for analysis of cellular function, providing a holistic view of the cellular processes to augment conventional labelled and omics approaches. “